# Microbiota-Driven Mechanisms in Multiple Sclerosis: Pathogenesis, Therapeutic Strategies, and Biomarker Potential

**DOI:** 10.3390/biology14040435

**Published:** 2025-04-17

**Authors:** Mohammad Hosein Nemati, Esmaeil Yazdanpanah, Roya Kazemi, Niloufar Orooji, Sepehr Dadfar, Valentyn Oksenych, Dariush Haghmorad

**Affiliations:** 1Student Research Committee, Semnan University of Medical Sciences, Semnan 3514799442, Iran; 2Department of Immunology, School of Medicine, Semnan University of Medical Sciences, Semnan 3514799442, Iran; 3Immunology Research Center, Mashhad University of Medical Sciences, Mashhad 9177948564, Iran; 4Broegelmann Research Laboratory, Department of Clinical Science, University of Bergen, 5020 Bergen, Norway

**Keywords:** gut microbiota, multiple sclerosis, immune modulation, dysbiosis, therapeutic interventions

## Abstract

Multiple sclerosis is a long-term disease that affects the brain and spinal cord, leading to problems with movement, balance, and thinking. Recent research suggests that the bacteria living in our gut play a key role in how the immune system responds to this disease. An imbalance in these gut bacteria can increase inflammation and worsen symptoms. In this review, we explore how gut bacteria influence immune cells and how certain bacterial species may help either protect against or contribute to disease progression. We also discuss potential treatments that target gut bacteria, such as probiotics, dietary changes, and fecal transplants, which have shown promise in improving symptoms. Additionally, we highlight how changes in gut bacteria could serve as early warning signs for disease diagnosis and treatment response. Understanding these connections may lead to better, more personalized treatments for people with multiple sclerosis and other related diseases. This knowledge could help improve quality of life and provide new strategies for managing the condition more effectively.

## 1. The Role of Gut Microbiota in the Pathogenesis of Multiple Sclerosis

### 1.1. Overview of the Gut–Brain Axis

Over recent years, the role of the gut microbiome in human health and disease has sparked increasing attention. It has become a topic of great scientific and public interest, especially with new terms such as gut–mouth axis, gut–immunity axis, gut–brain axis, gut–skin axis, and other yet-to-be-discovered relationships [1]. The human microbiota, especially the gut microbiota, has even been considered an “essential organ” due to its significant genetic contribution, carrying about 150-fold more genes than the entire human genome [2]. The mammalian microbiota is estimated to contain trillions of symbiotic microorganisms, comprising bacteria, archaea, viruses, and eukaryotes, residing internally and externally in the body [3].

Most members of the human microbiota reside in the gastrointestinal tract (more than 97%), especially in the colon, and are commonly referred to as the gut microbiota [4]. The microbiota is usually defined as the assemblage of microorganisms living symbiotically on and within the human body, whereas the collective genome of these microorganisms is referred to as the microbiome [5]. In this regard, the microbiome encompasses a broader spectrum than the microbiota [3]. The most abundant bacteria in the human gut belong to Firmicutes, Bacteroidetes, Actinobacteria, Proteobacteria, and Verrucomicrobia, accounting for more than 98% of the gut microbiota [6]. Its composition is shaped by a range of host-related factors, including geographical origin, genetic predisposition, and early microbial exposure at birth. However, environmental influences, particularly dietary patterns (e.g., high sugar intake and low fiber consumption), exposure to xenobiotics (such as antibiotics, pharmaceuticals, and food additives), and hygiene practices, play a more significant role in modulating its structure and function [7,8].

A growing body of evidence from both preclinical and clinical research has firmly established the essential role of the gut microbiota in maintaining and promoting human health. Beyond its direct impact on the gastrointestinal tract, the gut microbiota also exerts significant influence on distant organs, including the brain, liver, and pancreas, through complex neuroendocrine and metabolic signaling pathways [9]. It profoundly influences many aspects of human physiology, such as metabolism, protection against pathogens, epithelial cell proliferation and differentiation, maintaining gut barrier integrity, nutrient extraction, vitamin production, immune system maturation, and immune homeostasis [3,8].

In recent years, numerous researchers have focused on understanding how the gastrointestinal tract can affect the brain. This connection, termed the gut–brain axis, refers to bidirectional biochemical communication between the central nervous system (CNS) and the gastrointestinal tract. The enteric neural system facilitates the flow of information in both directions [10]. Accumulating evidence points to the critical role of the gut microbiota in regulating the brain, anxiety, stress, depressive symptoms, and social behavior [2]. Although the nervous system and gut are anatomically separated, several pathways by which the gut microbiota interacts with the CNS have been suggested [11]. The gut–brain axis is carried out by the physiological channels, which encompass the CNS, the autonomic nervous system (ANS), and the enteric nervous system (ENS), as well as the neuroendocrine and neuroimmune pathways and signaling molecules produced by gut microbes [12].

The gut microbiota communicates with the CNS through at least three parallel and interacting channels involving neuronal, endocrine, and immune signaling channels [13,14]. For example, the gut microbiota is capable of producing and consuming neurotransmitters such as glutamate, dopamine, gamma-aminobutyric acid (GABA), serotonin (5-HT), and norepinephrine, which can participate in the gut–brain axis pathway and reach the CNS to modulate the functions of astrocytes, neurons, and the BBB [11,15]. In turn, the CNS, particularly the brain, modulates gut function through the physiological control of neurons, hormones, and cytokines [13,14]. Evidence indicates that microbiota communication with the brain involves the vagus nerve, which transmits information from the luminal environment to the CNS [10].

It is becoming increasingly clear that dysbiosis alterations in the composition and/or functions of the microbiome are implicated not only in gastrointestinal disorders but also in various pathological conditions, including inflammatory bowel disease (IBD), cardiovascular disease, obesity, depression, autism, hypertension, diabetes, allergy, autoimmune diseases, neurodegenerative diseases, and cancers [8,16,17].

Recent advances in gut microbiota research have significantly expanded our understanding of its role in multiple sclerosis (MS), necessitating an updated synthesis of the latest findings. While previous reviews have explored microbiota–immune system interactions, this review uniquely integrates emerging insights into the immunomodulatory effects of gut microbiota, emphasizing its role in shaping T-cell responses and metabolic pathways relevant to MS pathogenesis. Additionally, we provide a comprehensive discussion on microbiota-targeted therapeutic strategies, including probiotics, prebiotics, fecal microbiota transplantation (FMT), and dietary interventions, highlighting their clinical relevance and potential for personalized treatment approaches. Moreover, we explore the biomarker potential of gut microbiota, examining its utility in MS diagnosis, disease progression monitoring, and treatment response assessment.

To ensure a comprehensive and systematic approach, we performed an extensive literature search across multiple databases, including PubMed, Scopus, and Web of Science. The search strategy incorporated key terms such as “gut microbiota AND multiple sclerosis”, “microbiota-targeted therapies AND MS”, and “microbiome AND neuroinflammation” to capture relevant studies published up to December 2024. Articles were selected based on predefined inclusion criteria, including peer-reviewed studies that focused on the gut microbiota’s role in MS pathogenesis, immune modulation, therapeutic interventions, and biomarker discovery. Exclusion criteria included non-English articles, studies with insufficient microbiota-related data, and reviews that did not provide novel insights.

### 1.2. Immunomodulatory Effects of Microbiota

The gut microbiota was first identified as a commensal organism involved in food digestion. However, growing evidence from human and animal studies supports its critical role in the physiological and pathological immune systems [18]. As mentioned above, the largest concentrations of microbes and the body’s immune system lie within the gastrointestinal tract, where the gut microbiota and host immune system act symbiotically [19]. Bidirectional cross-talk between the commensal microbiota and the immune system is essential to developing and modulating host immune responses and forming immunological memory [20].

#### 1.2.1. Immunomodulatory Effects on T Cells

A crucial step toward understanding the importance of the microbiome in shaping the enteric immune system was the study performed in animals raised in the absence of live microbes, referred to as germ-free (GF), which allowed researchers to study immune responses in the absence of a microbiota. As a go-to animal model for bacteria–host interactions, GF animals exhibited immature intestinal lymphoid tissues, decreased serum immunoglobulin levels, a lack of T-helper (Th) 17 cells, a reduction in regulatory T cells (Treg cells), and an imbalance in Th1/Th2 cells, which is skewed towards the Th2 response [6,18,21,22,23]. However, mice with a naturally complex microbiome tend to have more significant macrophage digestive ability and more rapid immune response, indicating the vital role of the gut microbiota in promoting immune system maturation [6,19].

The gut microbiome plays a pivotal role in the initiation, modulation, and functionality of key components of the host’s innate and adaptive immune system, contributing to immune development, regulation, and homeostasis [19,24]. The innate immune system recognizes the colonization pattern of gut microbiota via the interactions of microbe-associated molecular patterns (MAMPs) and pattern recognition receptors (PRRs) expressed in most immune cells, which trigger the activation of the adaptive immune response to recognize and eliminate pathogens or promote immunotolerance [25]. Furthermore, commensal bacteria can provide several signals leading to the differentiation of T-cell phenotypes into four major cell types: GATA3^+^ Th2 cells, T-bet^+^ Th1 cells, RORγt^+^ Th17 cells, or into T cells with a regulatory phenotype (FOXP3^+^ Tregs), as well as modulating Th17/Treg cell balance (Figure 1) [26,27].

One important recent finding is that numerous members of the microbiota have a profound effect on the induction of Treg cells in both the small intestinal and colonic lamina propria, including *Escherichia*, *Akkermansia*, *bifidobacteria*, *Bacteroides*, *Clostridium*, *Lactobacillus*, and *Streptococcus* strains [11,28,29]. The capsular polysaccharide A derived from human commensal *Bacteroides fragilis* influences T-cell activation by engaging with Toll-like receptor 2 (TLR2) expressed by T cells. It actively restrains Th17 differentiation and favors the anti-inflammatory function of Tregs, leading to the maintenance of immune homeostasis [30]. Moreover, some Lactobacillus strains can upregulate Treg cell induction, suppress Th1 and Th17, alter the Th1/Th2 ratio, and influence the subset ratio of M1/M2 macrophages [31].

Unlike most gut microbes, specific members of the commensal microbiota known as segmented filamentous bacteria (SFB), with the candidate name Arthromitus, were shown to be a specific inducer of Th17 cell differentiation in the lamina propria [32,33]. Upon direct adherence to intestinal epithelial cells, SFB induced the production of serum amyloid A (SAA), which acts on lamina propria dendritic cells (LP DCs) to induce the production of IL-1β, IL-6, IL-23, and reactive oxygen species (ROS), which in turn foster differentiation of naive CD4^+^ T cells into Th17 cells. Thus, SFB colonization resulted in reduced colonization and the growth of pathogenic bacteria [33].

As further evidence, increased *Prevotella* abundance in inflammatory disorders is consistent with Th17-mediated mucosal inflammation through the production of IL-1β, IL-6, and IL-23 by DC in a TLR2-dependent manner [34,35]. Several other mucosal-associated microbiota members, such as *Escherichia coli*, *Bifidobacterium adolescentis*, *Staphylococcus aureus*, and *Candida albicans*, have been shown to contribute to the development of inflammatory Th17 cells [36]. Therefore, commensal gut microbiota imparts both pro- and anti-inflammatory signals and shapes the host T-cell populations. Consequently, disturbances in the intestinal microbial population are associated with the propagation of pro-inflammatory signals, an increase in the Th17/Treg ratio, and a loss of self-tolerance [23,37].

In addition to affecting cellular immunity, another point stressed by another study is the role of gut microbiota metabolites and antigens as a triggering factor in the development and function of humoral immunity (Figure 1) [38].

#### 1.2.2. Immunomodulatory Effects on B Cells

Studies have demonstrated that resident intestinal commensals, including bacterial antigens and metabolites, profoundly influence many aspects of B cells, including their differentiation, development, function, and activation [38]. Studies with GF animals reveal that the microbiota intestinal bacteria are required for the development of gut-associated lymphoid tissues (GALTs), the largest immune organ in the body, and promote mucosal barrier function [39]. The production of IgA by conventional B cells in the intestinal mucosa is one of the immune responses following bacterial intestinal colonization [38,40]. Following this fact, the production of IgA is impaired in germ-free animals [41], and the intestinal colonization of segmented filamentous bacteria (SFB), E. coli, and *bifidobacteria* induces B-cell differentiation and activation and enhances the specific IgA antibody response [42,43].

IgA is one of the most important and dominant immunoglobulin classes, regulating colonization, invasion, and growth, and maintaining gut microbiota homeostasis [37,44]. Intestinal IgA plays an important role in controlling pathogens and toxins, including preventing pathogen adhesion in the gut and penetration into the intestinal barrier, agglutinating pathogens, and facilitating their clearance and clearance of bacterial toxin by combining with them [41]. On the other hand, the differentiation of naïve B cells into regulatory B cells is triggered by gut microbial metabolites in the periphery and in gut-associated lymphoid tissues (GALTs) [45]. In addition, gut microbiota depletion leads to decreased B regulatory cell (Breg) numbers in the spleen and reduced circulating IL-10 and IL-35 (Figure 1) [46].

Notably, numerous metabolites produced by commensal bacteria, including short-chain fatty acids, tryptophan metabolites, and bile acid derivatives, have been identified as key regulators in the development, homeostasis, and function of the immune system [47,48]. Among these microbiota-derived metabolites, short-chain fatty acids (SCFAs) have drawn a lot of attention. SCFAs, namely butyric acid, acetic acid, and propionic acid, are organic acids produced by the bacterial fermentation of undigested carbohydrates in the colonic lumen [49]. Many studies have shown that not only innate immune cells but also adaptive immune cells are modulated by SCFAs [50,51]. In addition to providing energy for the host, SCFAs also exert anti-inflammatory and immunomodulatory effects mediated by increasing anti-inflammatory Treg cells and decreasing pro-inflammatory T cells (Th1 and Th17), which lead to an anti-inflammatory response state [6,18]. In addition, it is reported that SCFAs favor the production of anti-inflammatory cytokines such as IL-10 and TGF-β and inhibit the production of proinflammatory molecules, including IFN-γ, TNF-α, IL-6, IL-1β, and IL-8 [52].

### 1.3. Key Microbial Players

“Microbiome” and “autoimmunity” are the most interesting scientific research topics of today’s modern age. Although the specific relationship between microbiota and MS remains to be elucidated, experimental and clinical studies provide compelling evidence that supports the potential role of the gut microbiota in the pathogenesis of MS [15]. Accordingly, given that specific pathogen-free (SPF) animals are more susceptible to experimental autoimmune encephalomyelitis (EAE) than germ-free and antibiotic-treated mice, this highlights the importance of the microbiota in MS initiation and severity [53,54]. Furthermore, given that gut microbiota-derived signals are essential for the growth and maturation of the immune system, it is reasonable to anticipate their involvement in the pathogenesis and progression of MS [55]. Recently, with the revolutionary advances in next-generation sequencing (NGS) techniques, numerous reports in human and murine models underscore the association of dysbiosis of the gut microbiota with inflammatory and autoimmune diseases such as type 1 diabetes, SLE, RA, and MS [56,57].

Recently, studies have identified a strong association between specific microbial taxa and the development of both MS and its animal model, EAE [58]. Supportive of this postulate, the microbiome signature in MS patients is characterized by a reduction in microbiota diversity, with a significantly low abundance of *F. prausnitzii*, Prevotella, Lactobacillus, and Bacteroides and a higher abundance of *Akkermansia muciniphila* and Acinetobacter [56,59]. In another exploratory study of 31 RRMS patients, certain taxa, such as Psuedomonas, Mycoplana, Haemophilus, Blautia, and Dorea genera, were increased, whereas the levels of Parabacteroides, Adlercreutzia, and Prevotella were decreased in MS [60].

Growing empirical evidence suggests that decreased gut microbiota diversity and alterations in specific taxa are associated with an imbalance between pro- and anti-inflammatory immune responses and subsequent Treg/Th17/Th1 imbalance, which correlates with the onset and/or progression of the disease of MS [54,61]. Subsequent studies confirmed that some microbial strains, such as Anaeroplasma, Rikenellaceae, and Clostridium, were positively related to disease severity, whereas Bifidobacterium, Prevotella, and Lactobacillus were negatively related to EAE severity, which suggests the potential application of these variations for predicting MS disease status [18]. The results showed that alterations in the compositional diversity and abundance levels of microbiota correlated with the disease onset and clinical severity of EAE [18,23].

The increased abundance of *Akkermansia muciniphila* and *Acinetobacter calcoaceticus* in patients with MS aligns with previous findings, which have demonstrated their ability to induce proinflammatory responses in human peripheral blood mononuclear cells and in monocolonized mouse models [18,62]. Other studies have shown that monocolonization of the gut in C57BL/6 mice with segmented filamentous bacteria (SFB) exacerbates disease severity by stimulating IL-17 production in the gut and enhancing IL-17A–producing CD4^+^ T cells (Th17) in the CNS. These findings provide compelling evidence for a link between immune cell activation in the gut and the onset of neurological inflammation [61]. Another resident of the human gut microbiome influencing T-cell homeostasis is *Akkermansia muciniphila*, playing an important role in the pathogenesis of MS through Th1 differentiation [62]. A higher Firmicutes/Bacteroidetes ratio and increased Streptococcus concentration may contribute to the exacerbation of chronic inflammation and the exacerbation of MS symptoms by shifting the immune response to the Th17 phenotype [55].

In contrast, colonization of C57BL/6 mice with purified polysaccharide A (PSA), a capsular component of the gut commensal Bacteroides fragilis, conferred both prophylactic and therapeutic protection against the development of EAE. This protective effect was attributed to the induction of anti-inflammatory Treg activity and the modulation of pro-inflammatory Th1 responses [63]. Other studies have indicated that treatment with a human commensal, *Prevotella histocola*, provided protection against EAE, as it induced the number of CD4^+^FoxP3^+^ Tregs and tolerogenic dendritic cells while reducing pro-inflammatory Th1 and Th17 cells and suppressing macrophages [64,65].

Notably, treatment with the gut commensal bacterium Prevotella histicola suppresses CNS inflammatory and demyelinating disease as efficiently as the MS therapies glatiramer acetate and interferon beta by modulating systemic immune responses, including downregulation of pro-inflammatory Th1/Th17 responses, while increasing CD4^+^FoxP3^+^ Treg cells, tolerogenic DCs, and suppressive macrophages [66]. Subsequent work identified commensal *Parabacteroides distasonis* and Clostridia as protective bacteria that promote protection against inflammatory CNS demyelination by enhancing Tregs [67,68,69].

Short-chain fatty acids, among the most abundant metabolites produced by the gut microbiota, possess significant immunomodulatory properties. They exert their effects by suppressing pro-inflammatory Th1 and Th17 responses while enhancing the population of Tregs. Additionally, SCFAs contribute to maintaining the balance between Tregs and effector T cells, thereby promoting an overall anti-inflammatory immune state [70]. Therefore, gut dysbiosis leads to an altered balance between long-chain fatty acids with pro-inflammatory properties and SCFAs with strong immunoregulatory and anti-inflammatory properties, and has been implicated in the development of EAE [66,71]. These findings suggest that significant alterations in the microbial taxa of the gut microbiome and subsequent SCFA production deficiency in the gut are at least partially associated with lowering the threshold for the development and exacerbation of MS [72,73].

Taken together, data from MS patients and the murine EAE model suggest that the altered composition and functionality of the microbiota, known as dysbiosis, can contribute to CNS-specific autoimmunity through the imbalance of T-cell subpopulations such as Th1, Th2, Th17, and Treg cells. Therefore, restoring the diversity with probiotic treatment could be considered a novel biotherapeutic approach in the prevention and treatment of MS.

## 2. Alterations in Gut Microbiota Composition in Multiple Sclerosis Patients

### 2.1. Microbiome Profiling

The microbiome comprises millions of microorganisms capable of influencing neurological diseases by modulating immune cells that migrate from the gut to the brain. The gut microbiota exerts its effects on the CNS through three primary pathways: neurotransmitter signaling, neuroendocrine regulation, and neuroimmune interactions [74]. Studies in animal models have demonstrated that the gut microbiota plays a crucial role in neuroinflammation. Notably, germ-free and antibiotic-treated mice exhibit resistance to both induced and spontaneous EAE, the widely used murine model for MS, highlighting the microbiota’s involvement in disease pathogenesis [75,76]. Several studies suggest a correlation between gut microbiota and multiple sclerosis (MS), indicating that broad-spectrum antibiotic treatment is associated with a slower progression of EAE. This effect is accompanied by a reduction in proinflammatory cytokine levels and a decrease in mesenteric Th17 cell populations, further supporting the role of gut microbiota in modulating neuroinflammation [75].

Moreover, the gut microbiota plays a crucial role in regulating the permeability of the blood–brain barrier (BBB), potentially impacting central nervous system homeostasis and neuroinflammatory processes [77]. The administration of bacteria that promote Treg induction, such as Bacteroides fragilis expressing PSA, has been shown to alleviate EAE, highlighting the therapeutic potential of gut microbiota modulation in neuroinflammatory diseases [63]. In contrast, bacteria that stimulate Th17 cell responses can exacerbate EAE. Recent studies analyzing the gut microbiota composition in MS patients have revealed a reduction in beneficial microbial species alongside an increase in proinflammatory bacteria associated with the regulation of autoimmunity, further implicating gut dysbiosis in MS pathogenesis [78,79,80]. These findings highlight the pivotal role of gut microbiota in the pathogenesis of MS and its potential as a target for therapeutic intervention.

### 2.2. Disease Phenotypes and Microbiota

Several studies have investigated the differences in gut microbiota composition between MS patients and healthy controls, identifying a state of dysbiosis in MS. This dysbiosis is marked by both a depletion of beneficial microbial taxa and an enrichment in proinflammatory species, suggesting a potential role in disease pathogenesis. Chen et al. observed increases in Pseudomonas, Mycoplasma, Haemophilus, Blautia, and Dorea genera, and decreases in Parabacteroides, Adlercreutzia, and Prevotella genera in relapsing–remitting MS (RRMS) patients [60,81]. Jangi et al. found increased levels of Methanobrevibacter and Akkermansia, while Butyricimonas was reduced in RRMS patients. Additionally, these microbial alterations were correlated with changes in gene expression linked to dendritic cell maturation, interferon signaling, and NF-κB signaling pathways in circulating T cells and monocytes, further implicating gut microbiota in the modulation of immune responses in MS [79].

The influence of MS drug treatment on gut microbiota composition has also been highlighted. MS patients undergoing treatment with interferon β or glatiramer acetate exhibited an increased abundance of Prevotella and Sutterella, along with a reduction in Sarcina, compared to untreated individuals, suggesting a potential interaction between immunomodulatory therapy and gut microbial dynamics [79]. Another study showed that vitamin D administration in RRMS patients resulted in decreased Bacteroidaceae and Faecalibacterium levels, and an increase in Ruminococcus [82]. A Japanese study found that several gut bacteria belonging to Clostridia clusters XIVa and IV were significantly reduced in MS patients [78]. Tremlett et al. investigated the association between gut microbiota diversity and relapse risk in pediatric MS, finding that a reduction in Fusobacteria was linked to an increased risk of relapse. This suggests a potential role for specific microbial taxa in disease progression and relapse susceptibility [83,84].

Furthermore, an analysis of small-intestinal tissues from MS patients during the active disease phase revealed an increased Firmicutes/Bacteroidetes ratio, along with a higher abundance of Streptococcus and a reduction in Prevotella compared to both healthy controls and patients in remission. Cosorich et al. reported that the proportion of Firmicutes to Bacteroidetes and the presence of Streptococcus were elevated, whereas Prevotella abundance was diminished in patients with active MS relative to healthy individuals and those in remission, further implicating gut microbiota alterations in disease activity [85].

These studies demonstrate how microbiota composition may influence MS disease phenotypes and suggest that microbiota modulation could have therapeutic potential in managing MS.

## 3. Therapeutic Potential of Modulating Gut Microbiota in Multiple Sclerosis

The potential of gut microbiota modulation as a therapeutic strategy for MS has attracted considerable interest in recent years. Increasing evidence highlights the intricate interplay between the gut microbiome and immune system function, suggesting that targeted microbial interventions could influence disease progression and treatment outcomes. Dysbiosis, or an imbalance in gut microbial communities, has been associated with the development of MS, potentially impacting inflammatory responses and neurodegeneration. Restoring a balanced gut microbiota through dietary modifications, probiotic supplementation, or fecal microbiota transplantation holds promise for improving gut–brain axis communication, mitigating neuroinflammation, and promoting regulatory immune responses, potentially offering novel therapeutic approaches for MS. This approach not only holds promise for alleviating symptoms and delaying disease progression but also represents a new avenue for personalized MS treatment, emphasizing the gut microbiome as a target for future therapeutic development.

### 3.1. Probiotics and Prebiotics

Prebiotics are dietary fibers or foods that promote the growth and/or activity of beneficial indigenous probiotic bacteria [86]. They are resistant to the adverse effects of gastric acid and digestive enzymes [87]. Numerous prebiotics with varying origins and chemical properties exist, but the most prevalent are fructans (fructo-oligosaccharides (FOSs) and inulin) and galacto-oligosaccharides (GOSs) [88]. Prebiotic-rich foods include soybeans, raw oats, unrefined wheat, barley, yacon, nondigestible polysaccharides, and inulin-type fructans derived from traditional Chinese medicine (TCM) [89].

Fructans consist of a linear chain of fructose with a β (2 → 1) bond. The degree of polymerization (DP) of inulin is approximately 60, whereas FOSs have a DP of less than 10. Fructans preferentially activate lactic acid bacteria, but recent research shows that the chain length of fructans plays a key role in selecting bacteria for fermentation, thereby stimulating other bacterial species either directly or indirectly [90,91]. GOSs, resulting from lactose extension, are classified into two subgroups: one with extra galactose at C3, C4, or C6, and another produced through enzymatic trans-glycosylation from lactose, referred to as trans-galacto-oligosaccharides (TOSs). GOSs significantly increase the activity and/or growth of Bifidobacteria and Lactobacilli, with a smaller effect on other Bacteroidetes and Firmicutes species [90,92,93].

Other types of prebiotics include resistant starch and non-carbohydrate substances such as flavonoids. Resistant starch has therapeutic benefits due to the production of high quantities of butyrate, selectively benefiting Firmicutes [94,95,96]. Flavonoids have been shown to stimulate lactic acid bacteria both in vitro and in vivo [97]. Ultimately, prebiotics influence the composition and function of the gut microbiota, allowing probiotic bacteria to survive and proliferate during their transit through the upper gastrointestinal tract (GIT), with minimal interference from other microbes [98].

In an animal model, dietary non-fermentable fiber, which inhibits EAE, was found to alter gut microbiota composition (reducing diversity and increasing the abundance of Ruminococcaceae, Helicobacteraceae, and Enterococcaceae) and metabolic profiles, stimulating suppressive Th2 responses [99]. Furthermore, a clinical study found a correlation between the dietary fiber intake of MS patients and anthropometric measurements, disability levels, and markers of systemic inflammation [100].

Probiotics have been extensively studied as potential treatments for microbiota-related diseases, demonstrating a range of immunoregulatory activities, such as immune response modulation, pathogen colonization inhibition, intestinal microbial homeostasis regulation, and preservation of the gastrointestinal barrier [101]. Probiotic administration has been shown to help manage EAE by lowering incidence, delaying symptom onset, and reducing symptoms [102].

Studies on animal models revealed beneficial effects of oral or intraperitoneal administration of Lactobacillus, Escherichia coli, and Prevotella strains, which either prevented EAE onset or alleviated its progression [102]. Marmoset twins were fed yogurt-based (YBD) or water-based (WBD) diets before immunization. The YBD group exhibited less demyelination and a lower pro-inflammatory response among T cells, B cells, and cytokines, with some individuals showing no symptoms of EAE. Changes in the gut microbiome were observed only in the YBD group following immunization, suggesting a relationship between diet and immune responses [103].

Contradictory findings have been reported regarding Lactobacillus reuteri strains, which may exacerbate EAE due to their interaction with other bacteria and activation of molecular mimicry [102]. Tankou et al. studied the effects of a probiotic combination (Lactobacillus, Bifidobacterium, and Streptococcus) on MS patients and healthy individuals for two months, finding significant shifts in microbiota composition, with an increase in probiotic species and a reduction in α-diversity. Additionally, probiotic administration reduced CD14 and CD80 expression on peripheral monocytes [104].

In EAE models, engineered probiotics expressing heat shock proteins and elafin demonstrated anti-inflammatory properties [105]. The immunomodulatory effects of Tregs, IL-35, IL-10, neurotransmitters, SCFAs, and other therapeutic compounds identified in human and animal studies provide novel therapeutic targets for genetically engineered probiotics [106].

### 3.2. Fecal Microbiota Transplantation (FMT)

Fecal microbiota transplantation (FMT) involves replacing the entire gut microbiome to address deficiencies in its structure and function. The diseased state is reversed by removing the abnormal gut microbiota and replacing it with a healthy one. FMT has shown promise in treating various diseases, including neurological disorders and autoimmune diseases such as inflammatory bowel disease (IBD), which are thought to be linked to gut dysbiosis [54]. Recently, FMT has garnered attention for its potential to address disorders related to gut microbiota. It may benefit MS patients in the same way it has aided those with non-alcoholic fatty liver disease by improving intestinal permeability [107]. In MS patients, the gut microbiota contains fewer microbial species that induce Tregs, leading to a higher presence of peripheral Th1/Th17 cells. This imbalance results in CNS inflammation and increased BBB permeability, exacerbating brain inflammation. Modifying the microbiota to promote Treg induction may help reduce pathogenic T cells [15].

Animal studies have shown that FMT reduces Akkermansia genus abundance while increasing Prevotella genus levels, mirroring the effects of probiotic therapy [104]. Few studies have explored the impact of FMT on MS. One preliminary abstract reported positive effects of FMT on the neurological disorders of three MS patients [108]. Another study reported that FMT helped three MS patients with severe constipation return to normal bowel function, along with a significant improvement in their exercise capacity [108]. A case report provided the first evidence of FMT’s effects on a patients with secondary progressive MS (SPMS), where EDSS was stabilized over ten years, indicating potential long-term benefits [109].

In MS experimental models, FMT from healthy mice to immunized mice altered the gut microbiota composition, reducing EAE symptom intensity, neurodegenerative marker expression, and pathological symptoms [110]. A study by Berer et al. involved colonizing SJL mice with gut microbiota from discordant human twins to diagnose MS [80]. More recently, a single-subject study following an MS patients for a year after double FMT provided evidence for FMT’s positive effects, including gait improvement linked to microbial changes and higher levels of brain-derived neurotrophic factor [111].

FMT in a randomized controlled trial of RRMS individuals led to donor-specific changes in microbiome composition, though large intraindividual variability prevented significant species diversity changes [112]. Despite early termination, the trial indicated FMT’s potential to modify gut microbiome composition. Further research is needed to validate these results. Laeeq et al. found that FMT relieved neurological symptoms in 15 MS patients, with lasting improvements and no adverse effects, suggesting FMT as a promising MS treatment [113]. FMT enhanced neurological function, lowered IL-17 levels, and restored intestinal homeostasis, suggesting the efferent activity of the hypothalamic–pituitary–adrenal axis as a contributing factor [114]. However, large-scale investigations are required to fully explore FMT’s therapeutic potential for MS.

### 3.3. Dietary Interventions

The primary factor influencing the composition of the gut microbiome is diet. The structure and function of the gut microbiome can be modified by adjusting the substances and nutrients to which it is exposed [115]. Dietary factors regulate the ratio of pro-inflammatory to anti-inflammatory responses, potentially influencing autoimmune diseases, including MS [116]. However, due to variations in study designs and variables examined, consistent and compelling evidence of the efficacy of dietary interventions in MS patients remains limited [117].

Dietary therapies for MS have been extensively studied for their various interactions with gut flora. Certain food substances, acting as metabolic substrates, may promote the growth of specific bacterial strains while inhibiting others, thereby altering the microbial composition. Dietary influences on host–microbiota immunological interactions, such as immune sampling in intestinal Peyer’s patches, are examples of indirect impacts. Additionally, diet may affect the integrity and functionality of the intestinal barrier. These associations highlight the potential role of dietary interventions as part of a comprehensive therapeutic strategy for MS [118,119].

Key metabolites derived from dietary components that can interact with the immune and neurological systems include short-chain fatty acids, tryptophan, polyamines, and urolithins. Major SCFAs (acetate, propionate, and butyrate), urolithins, spermidine (a byproduct of L-arginine metabolism), and tryptophan metabolites (indole-3-lactic acid [ILA], indole-3-acetic acid [IAA], and indole-3-aldehyde [IAld]) are known to be produced by gut bacteria. SCFAs modulate Tregs by inhibiting histone deacetylase (HDAC), stimulating the transcription of IL-10 and FoxP3 genes, and activating free fatty acid receptor 2 (FFAR2)-dependent IL-10 production. ILA and urolithin inhibit Th17 polarization and reduce aryl hydrocarbon receptor (AhR)-mediated IL-17 production. Spermidine enhances arginase 1 (Arg-1) expression in macrophages and downregulates NF-κB pathway activity, limiting the release of pro-inflammatory cytokines and co-stimulatory molecules. SCFAs also cross the blood–brain barrier (BBB) via proton-dependent monocarboxylate transporter 1 (MCT1), restoring tight-junction proteins (occludin and claudin 5) and reducing reactive oxygen species (ROS) production in endothelial cells, thereby improving BBB integrity.

In the CNS, SCFAs inhibit pro-inflammatory microglia (M1) from producing TNF-α, IL-1β, IL-6, IL-12, and inducible nitric oxide synthase (iNOS). Similarly, urolithin A reduces pro-inflammatory markers (TNF-α, IL-1β, NO, PGE2). Tryptophan metabolites (e.g., indole-3-sulfate [I3S], indole-3-propionic acid [IPA], and IAld) suppress astrocyte-derived chemokines (MIP-1α, MCP-1, RANTES) in an AhR-dependent manner. Spermidine reduces astrocyte production of TNF-α, IL-6, and CCL2 (Figure 2) [120].

Studies on animal models have demonstrated that SCFAs, particularly acetate, reduce the severity of EAE in an IL-10-dependent manner [121]. Similar findings showed that colonization with microbiota from MS patients upregulated Treg-related gene expression when preceded by propionate therapy [122]. Preventive butyrate treatment decreased CNS inflammation and demyelination [123]. Polyunsaturated fatty acids (PUFAs) have shown protective effects in experimental MS models through immunomodulatory features that inhibit peripheral and CNS T-cell activity. PUFAs also enhance neuroprotection, promote remyelination, inhibit demyelination, and delay disease onset [124,125].

Research on polyphenols in EAE models has yielded mixed results. While flavonoids may reduce pro-inflammatory cytokines and alleviate symptoms, they can also delay recovery. Resveratrol demonstrated contradictory effects, worsening EAE in one study but promoting remyelination in another. Clinical evidence for polyphenols in MS is limited. Notably, nanocurcumin reduced pro-inflammatory cytokines, enhanced Treg activity, and improved quality of life for RRMS patients. However, further research is needed to explore polyphenols’ effects on the gut microbiome and their therapeutic potential [120].

The Western diet (WD), characterized by increased sodium chloride intake, influences tissue inflammation both locally and systemically, potentially altering gut microbiota composition and functionality. Elevated extracellular NaCl levels have been linked to increased Lachnospiraceae, Ruminococcus, and Prevotella spp. counts and reduced Lactobacillus levels in EAE models [126,127]. High NaCl levels also dysregulate immune homeostasis by preferentially activating pro-inflammatory Th17 cells and M1 macrophages while suppressing anti-inflammatory M2 macrophages and Tregs, particularly in the gut lamina propria [128]. Farez et al. associated a high-NaCl diet with increased demyelinating lesions on MRI and higher relapse rates [129]. However, other studies reported no significant effects of excessive NaCl intake on MS risk or relapse timing [130,131,132].

Fasting and ketogenic diets have shown positive effects in EAE models. Cignarella et al. demonstrated that fasting enriched specific gut microbiome families (e.g., Bacteroidaceae, Prevotellaceae, Lactobacillaceae) and promoted microbial antioxidant pathways. Fecal microbiota transplantation (FMT) from intermittently fasting mice improved EAE progression in mice on a normal diet, highlighting the gut microbiome’s role in CNS autoimmune conditions [133]. Swidsinski et al. investigated gut microbiome changes in MS patients during a 6-month ketogenic diet. Initial reductions in bacterial diversity were followed by recovery to baseline levels by week 12 and significant improvements by weeks 23–24, with notable increases in Akkermansia strains [134]. Current clinical investigations are expected to provide further insights into the interplay between the gut microbiome and dietary interventions in MS [135].

Cantoni et al. explored associations among the gut microbiome, immunological and metabolic parameters, diet, and clinical outcomes in MS patients. Changes in the gut microbiome and immunological dysregulation were observed in MS patients compared to healthy individuals, though these changes were unaffected by overall dietary composition. A correlation was found between meat consumption, elevated meat-related blood metabolites, reduced Bacteroides thetaiotaomicron (a polysaccharide-digesting bacterium), and increased Th17 cell levels [136]. Saresella et al. compared the effects of a high-vegetable/low-protein (HV/LP) diet and a WD on the gut microbiota and immune responses of MS patients. The HV/LP diet was associated with higher Lachnospiraceae abundance, reduced IL-17-producing T cells, and improved clinical outcomes, including lower relapse rates and disability levels [137].

Natural substances such as vitamin D significantly impact metabolic health. MS patients often exhibit lower vitamin D levels, potentially due to reduced sunlight exposure or other factors unrelated to geography, such as host metabolism and gut microbiota interactions [116]. Vitamin D supplementation may benefit MS patients by regulating the immune system, reducing intestinal permeability, and promoting the formation of immunomodulatory metabolites like butyrate [54]. However, further research is needed to establish optimal dosage and synergistic combinations with other supplements or medications [138]. A study on vitamin D supplementation in MS patients (untreated or taking glatiramer acetate) reported increased Faecalibacterium, Akkermansia, and Coprococcus genera in the untreated subgroup [82].

## 4. Microbiota-Driven Biomarkers for Multiple Sclerosis

In diagnosing MS, magnetic resonance imaging (MRI) is considered the gold standard. However, there is a growing need for reliable biomarkers for this condition. Among these, the neurofilament light chain (NfL) is currently being explored [139]. In MS, gut dysbiosis has been identified, which could serve as a marker for diagnosis, prognosis, and therapeutic outcomes. This is particularly relevant, as predicting long-term outcomes for MS patients remains challenging [3,139,140]. Below, we discuss studies that have investigated these aspects.

Navarro-López et al. conducted a study involving 15 patients with RRMS and 15 control subjects with similar dietary habits. The study identified gut dysbiosis characterized by alterations in several bacterial genera, including Lachnospiraceae, Ezakiella, Ruminococcaceae, Hungatella, Roseburia, Clostridium, Shuttleworthia, and Bilophila. Receiver operating characteristic (ROC) curves and area under the curve (AUC) analyses, with 95% confidence intervals (CIs), were calculated for these genera. The best predictive models were observed for Ezakiella (AUC: 75.0; CI: 60.6–89.4) and Bilophila (AUC: 70.2; CI: 50.1–90.4). These findings suggest that the presence of these bacteria in stool samples may serve as valuable predictive biomarkers for the diagnosis and prognosis of RRMS [140].

Devoldera et al. examined 111 patients by collecting fecal samples at baseline and three months later. The findings revealed that the Bacteroides 2 enterotype was predominantly associated with more severe disease presentations. Notably, this marker remained stable over three months, suggesting its potential as a robust prognostic indicator in MS, independent of confounding variables. Additionally, when compared to plasma levels of NfL, the Bacteroides 2 enterotype demonstrated a stronger correlation with MS-related disability. This stability and association underscore its value in monitoring disease progression and predicting patient outcomes [139].

Thirion et al. analyzed 148 Danish patients and 148 controls, revealing significant differences in 61 bacterial species. Among these, 31 species, including Clostridium leptum, Clostridium inocuum, Anaerotruncus colihominis, and Ruminococcus gnavus, were more prevalent in MS patients. Inflammatory markers, such as leukocytosis, C-reactive protein (CRP), and gene expression of IL-17A and IL-6, were positively correlated with these MS-associated species. Disease-active, treatment-naive cases showed direct associations between plasma cytokines such as IL-22, IL-17, IFN-β, IL-33, and TNF-α and these bacteria. Conversely, patients with non-active disease had higher levels of Faecalibacterium prausnitzii and Gordonibacter urolithinfaciens, which produce anti-inflammatory metabolites like butyrate and urolithin. Notably, serum propionic acid levels were lower in patients than in controls, with higher levels linked to improved symptoms, suggesting its therapeutic potential in MS management [66].

Zhou et al. studied 576 MS patients and 576 controls, finding significant increases in *Akkermansia muciniphila*, *Ruthenibacterium lactatiformans*, *Hungatella hathewayi*, and *Eisenbergiella tayi* in MS patients. At the same time, *Faecalibacterium prausnitzii* and Blautia species decreased, particularly during the relapsing-remitting phase. These microbiome shifts correlated with disease severity. The study also examined microbiome changes due to treatment, noting that fingolimod reduced *Bacteroides finegoldii*, *Roseburia faecis*, and Blautia species, while interferon-β treatment reduced Ruminococcus, Clostridium, and *Faecalibacterium prausnitzii* but increased *Parabacteroides distasonis*. These findings highlight the potential use of gut bacteria in assessing therapeutic responses, reinforcing the link between gut microbiota and the progression, prognosis, and treatment of MS [3].

Based on our literature review, alterations in gut microbiota—whether increases or decreases—could serve as diagnostic and prognostic markers and tools for monitoring treatment in conjunction with other diagnostic methods. Numerous studies have identified specific bacterial species that differ between MS patients and controls. Most notably, *Akkermansia muciniphila* was consistently elevated [141], while Prevotella species were frequently decreased [142]. A comprehensive summary of the bacteria reported to increase or decrease is provided in Table 1. However, the utility of these microbial markers has limitations, as factors such as the patient’s diet, geographic location, and stool sample collection methods can introduce variability [143].

## 5. The Interplay Between Gut Microbiota, Genetics, and Environmental Factors in Multiple Sclerosis

### 5.1. Gene–Microbiota Interactions

There are significant genetic risk factors associated with MS in patients who exhibit a different gut microbiota composition compared to healthy individuals [147]. Research indicates that variations in the MHC region can influence the shaping of the gut microbiome. This occurs through the selective colonization of certain bacterial species, due to either immune eradication mechanisms or their incapacity to adhere to the intestinal epithelium, often mediated by IgA selection [148,149]. Additionally, polymorphisms in HLA class II genes can affect disease progression by altering metabolic pathways and influencing the selection of bacteria within the gut microbiome [150].

Findings from experimental models reveal that the gut microbiota of HLA-DR3-expressing EAE-susceptible mice significantly differs from that of HLA-DR8-expressing mice resistant to the disease (Figure 3) [150].

Moreover, transferring the microbiota of genetically susceptible C57BL/6J (B6) mice to EAE-resistant PWD/PhJ (PWD) mice increases susceptibility to EAE, highlighting the interplay between the microbiome and the host genome [151].

Probiotic interventions demonstrate promise in mitigating myelin-degrading disorders such as MS by modifying gut microbiota composition. Digehsara et al. found that Lactobacillus species reduced inflammation in mice by downregulating inflammasome-associated genes, including NLRP-1, NLRP-3, and AIM2. Additionally, these bacteria suppressed the upregulation of the CYP27B1 gene in MS models (Figure 3) [152].

### 5.2. Environmental Influences

Obesity is a notable environmental risk factor that exacerbates MS severity. Shahi and colleagues demonstrated that high-fat diets in mice, which alter gut bacterial metabolic pathways—including increased sulfur, LPS, and long-chain fatty acid metabolism—led to increased intestinal permeability and the production of pro-inflammatory mediators such as MCP-1α and CCL-11. These mediators promote Th1 cell proliferation, aggravating the disease [153]. Their findings also revealed a higher prevalence of Proteobacteria and H2S-producing bacteria in the gut microbiota of EAE mice fed a high-fat diet compared to controls, correlating with increased inflammation.

Dietary composition profoundly affects the gut microbiota and its impact on MS. Diets rich in isoflavones, phytoestrogens found in legumes, reduced spinal cord injury severity and disease progression in EAE mice. These effects likely stem from insufficient activation of CD4^+^ T lymphocytes against myelin antigens. Additionally, the gut microbiota in these mice exhibited anti-inflammatory properties [154]. Switching from phytoestrogen-free diets to phytoestrogen-rich diets enhanced the diversity and stability of the gut microbiota and regulated LPS-mediated pro-inflammatory responses, providing protection for the central nervous system [155].

Antibiotics are another approach to modifying gut microbiota and alleviating EAE. Norfloxacin, a fluoroquinolone antibiotic, demonstrated immune-modulating effects by increasing Treg cell populations in the colon, spleen, lymph nodes, and CNS while reducing pro-inflammatory Th17 cells in EAE models [156].

### 5.3. Epigenetic Modifications

The gut microbiota is intricately linked to epigenetic modifications, including histone acetylation, DNA methylation, and non-coding RNA production [157]. By producing short-chain fatty acids (SCFAs) such as butyrate, the microbiota inhibits histone deacetylase (HDAC), enhancing intestinal anti-inflammatory responses and providing neuroprotection [157]. Multiple investigations have demonstrated that this inhibitory mechanism triggers the development of Treg cells, which defend the central nervous system (CNS) by controlling microglial cell activity and preserving the BBB [158,159,160].

Luu et al. identified that SCFAs, particularly pentanoate, inhibited specific HDAC enzymes, including HDAC8 and HDAC1. This inhibition suppressed IL-17A gene expression in Th17 cells, further illustrating the therapeutic potential of gut microbiota modulation [161].

## 6. Future Directions and Practical Applications of Gut Microbiota Research in Multiple Sclerosis

Investigation of the gut microbiota in MS appears to present an exciting interdisciplinary field for developing new therapeutic interventions. Although great progress has been made, many parts are still underexplored, and practical consequences from theory to clinical reality requires heavy investigation [54]. One of the most exciting applications currently being developed on gut microbiota research is personalized medicine therapy. Differences in gut microbiota composition among individuals suggest the need for personalized related interventions. Advanced sequencing technologies may characterize particular microbiota profiles for precise diagnosis and targeted therapies [162]. The discovery of dependable microbiota-derived biomarkers may contribute to prognosis, therapeutic response, and relapse prediction. The investigation of particular bacterial strains shown to have beneficial or adverse effects in MS (e.g., *Prevotella histicola* vs. *Akkermansia muciniphila*) has the potential to inform targeted probiotic/prebiotic formulations [163].

Expanding the therapeutic tools against MS by targeting the gut microbiota provides an innovative approach. By engineering probiotics (e.g., genetically modified with IL-10, SCFAs, and other anti-inflammatory molecules), it can manipulate adaptive immune responses. This could mean tailor-made prebiotics enriched beneficial bacteria such as *Lactobacillus* and *Bifidobacterium*, restoring microbial homeostasis and the gut–brain axis. These are preliminary studies that hint at an avenue, but standardized protocols and double-blinded trials are required to validate the utility of FMT in MS [164].

Given the significant role of diet in gut microbiota, dietary interventions could be a feasible and easily accessible approach to MS management. Studies indicate that ketogenic and anti-inflammatory diets have been associated with decreased gut pro-inflammatory bacteria over beneficial strains [118]. Dietary supplementation with vitamin D, omega-3 fatty acids, and polyphenol may interact synergistically with diet to enhance the microbiota profiles associated with reduced MS symptoms in humans. Local dietary habits and food-source adherence to nutritional recommendations can be improved for the sake of patient outcomes [165].

The application of multi-omics technologies (e.g., metagenomics, the transcriptome, and the metabolome) can reveal new insights into interactions between the gut microbiota and the host. What is needed are future studies that investigate the impact of CNS inflammation and neurodegeneration on microbial metabolites, and omics data integration for the development of predictive models for therapeutic responses and how common MS drugs (e.g., interferon-β) interact with gut microbiota and alter its composition [166].

While current evidence focusing on the gut microbiota targeted therapies, this field presents several open questions. To determine whether or not gut dysbiosis is a cause of MS and not just a consequence from the MS, longitudinal studies are necessary. Consensus for protocols of sampling, analysis, and therapeutic targeting of microbiota must be defined. Microbiota-based therapies should be demonstrated in large-scale randomized controlled trials (RCTs) to investigate sustainability and safety [15].

The gap between research and clinical practice can be closed by means of large and multicenter RCTs to confirm the efficacy of interventions targeting the microbiota. Patients and healthcare professionals need to be educated on the role of the gut microbiota in MS and the feasibility of therapeutic interventions. Collaboration with regulatory agencies will simplify the approval pathway for microbiota-based therapeutics [167,168].

The gut microbiota’s intricate relationship with MS presents unprecedented opportunities for innovative, patient-centered approaches. By addressing current limitations and advancing practical applications, future research can transform the understanding and management of MS, offering hope for improved patient outcomes.

### Limitations of EAE Models in MS Research

While EAE remains a widely used model for studying MS pathogenesis and therapeutic interventions, it has inherent limitations. EAE is induced through immunization with myelin antigens, such as myelin oligodendrocyte glycoprotein (MOG), which does not fully replicate the spontaneous onset of MS in human patients [169,170]. Additionally, EAE primarily models inflammatory demyelination but does not fully capture the neurodegenerative aspects of progressive MS. MS is influenced by a complex interplay of genetic predisposition and environmental triggers, including viral infections, gut microbiota composition, and vitamin D levels, which are not entirely represented in EAE models [171].

Given the complexity of MS, other complementary animal models have been developed to address specific disease aspects. The cuprizone model, which induces demyelination through toxic exposure, allows researchers to study remyelination and neurodegeneration independently of the immune system [172]. Similarly, the lysolecithin-induced demyelination model is used to investigate myelin repair mechanisms [173]. Additionally, viral infection models, such as the Theiler’s murine encephalomyelitis virus (TMEV) model, more accurately reflect the progressive and chronic neurodegenerative aspects of MS, including axonal loss and long-term CNS damage [174].

While each model has its own strengths and limitations, using a combination of different models can provide a more comprehensive understanding of MS pathophysiology. Future research should integrate alternative models and human-based studies to better mimic the complex genetic, environmental, and microbial factors contributing to MS, ultimately improving the translation of experimental findings into clinical applications.

## 7. Conclusions

Interactions of the gut microbiota and pathogenesis in MS pathogenesis further indicate the need for microbiome-based interventions in disease management. Dysbiosis is among the reasons for immune dysregulation and neuroinflammation, indicating that therapeutic strategies are needed to restore microbial equilibrium. The current evolution of probiotics, prebiotics, diet interventions, and fecal microbiota transplantation appears to have great potential in improving clinical outcomes and patient quality of life. Eventually, future research should attempt to target personalized therapeutic strategies based on biomarkers derived from microbiota that facilitate refinement to diagnosis and therapy. Addressing current challenges in microbiota research, such as standardization and long-term safety, will be pivotal in translating these findings into effective clinical practices.

## Figures and Tables

**Figure 1 biology-14-00435-f001:**
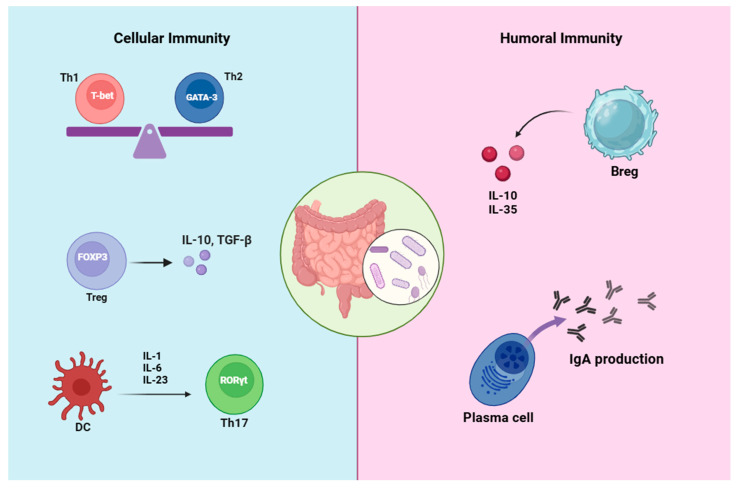
**Different combinations of gut commensal bacteria can generate signals that induce the differentiation of various T-cell subtypes.** Some of these bacteria cause dendritic cells to generate cytokines, including IL-1β, IL-6, and IL-23, which in turn causes Th17 cells to differentiate. Others can promote the production of Treg cells and affect the Th1/Th2 ratio, altering the composition of the intestinal T-cell population and establishing a balance between inflammatory and anti-inflammatory states in the gut. Bacterial metabolites and antigens present in the gut can influence naïve B cells, prompting their differentiation into Breg cells. These Breg cells produce anti-inflammatory cytokines such as IL-10, thereby inducing an anti-inflammatory environment within the gut. These bacteria also influence gut humoral immunity by inducing plasma cells to produce IgA antibodies, which play a crucial role in maintaining the homeostasis of gut bacteria.

**Figure 2 biology-14-00435-f002:**
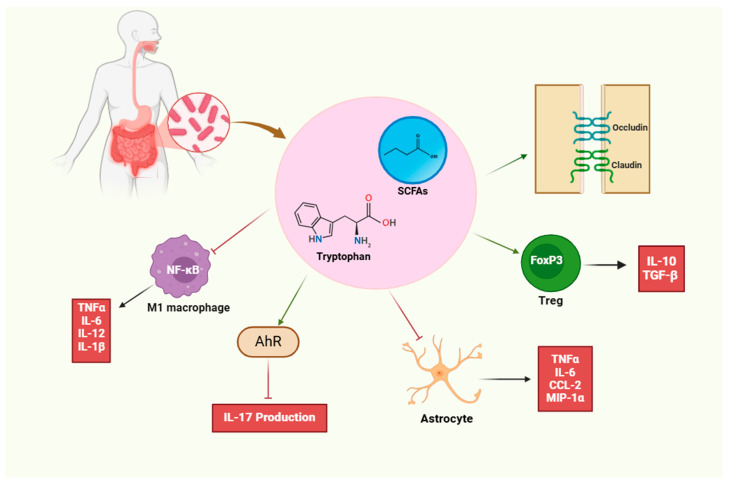
**Some food-derived metabolites that can influence the immune and nervous systems include short-chain fatty acids (SCFAs), the amino acid tryptophan, urolithins, and polyamines**. These compounds can inhibit inflammatory macrophages (M1) and astrocytes, preventing the production of pro-inflammatory cytokines such as TNF-α, IL-1β, IL-6, and CCL2. Conversely, by inducing the production of Treg cells and subsequently increasing anti-inflammatory cytokines like IL-10 and TGF-β, they contribute to the establishment of an anti-inflammatory environment. These metabolites, through the aryl hydrocarbon receptor (AhR), can reduce the production of the pro-inflammatory cytokine IL-17. On the other hand, according to the findings of some studies, these metabolites restore claudin and occludin proteins, thereby strengthening tight junctions and contributing to the maintenance of the blood–brain barrier’s integrity.

**Figure 3 biology-14-00435-f003:**
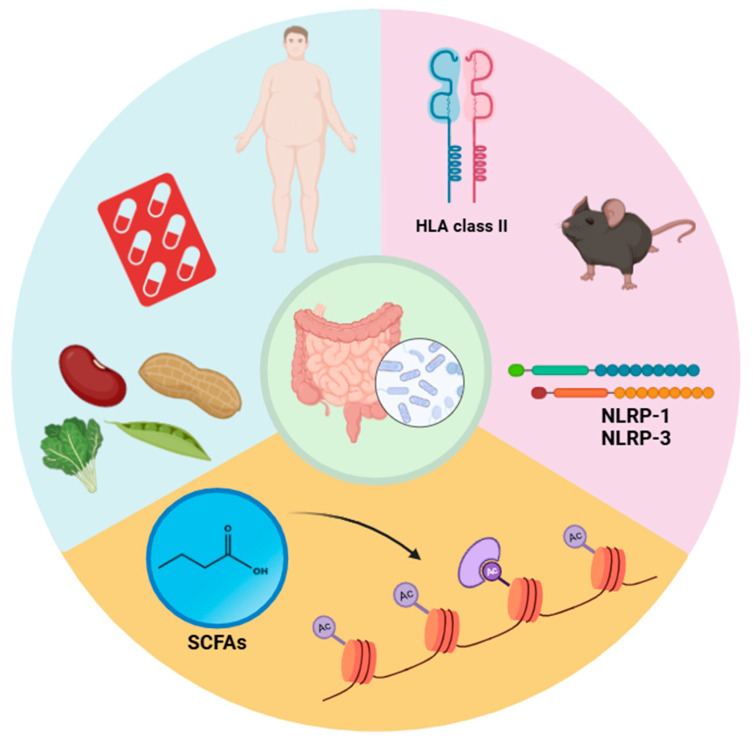
**Interaction between gut microbiota and genetics, epigenetics, and environmental agents in multiple sclerosis (MS).** Through a variety of mechanisms, the gut microbiota’s makeup can affect MS outcomes. As a significant environmental risk factor, obesity can alter the composition of the gut microbiome, leading to increased inflammation and disease aggravation. Different antibiotics and diets may also influence the gut microbiota, which can then affect the immune system and result in various outcomes for MS. The gut microorganisms of MS patients likewise are significantly influenced by host genetics. In particular, this association has been noted between the gut microbiota of these individuals and polymorphisms in the HLA class II gene. Additionally, a number of researchers have demonstrated the difference between the intestinal microbiomes of EAE-susceptible C57BL/6J mice and mice models that are resistant to the illness. The use of beneficial probiotics in the diet has also been associated with a lowering trend in the expression of genes linked to inflammation, such as NLRP-1 and NLRP-3. Gut bacteria’s production of short-chain fatty acids (SCFAs) might influence epigenetic modifications such as increased histone acetylation, which can promote the growth of Treg cells and preserve the blood–brain barrier.

**Table 1 biology-14-00435-t001:** **Microbiome dysbiosis in multiple sclerosis:** This table provides an overview of the changes in microbiome composition associated with multiple sclerosis (MS). It highlights specific microbial taxa, their classification, and references to relevant studies. The table outlines patterns of dysbiosis, including (↑) increases and (↓) decreases in various microbiota. The (↑↓) symbol indicates that taxa may be either increased or decreased and can potentially be used as biomarkers in this disease. These changes are compared to a control group or groups with better disease conditions. Additionally, PP MS refers to primary progressive multiple sclerosis.

MS Subtype	Microbial Taxa	Dysbiosis Pattern	References
RRMS	Lachnospiraceae	↑↓	[140,143]
	Ezakiella	↑↓	[140]
	Ruminococcaceae	↑↓	[140]
	Hungatella	↑↓	[140]
	*Hungatella effluvia*	↑	[66]
	*Hungatella hathewayi*	↑	[3]
	Roseburia	↑↓	[140,143]
	*Clostridium leptum*	↑	[66]
	*Clostridium innocuum*	↑	[66]
	Shuttleworthia	↑↓	[140]
	*Bilophila wadsworthia*	↑	[66]
	Prevotella	↓	[85,143,144]
	Streptococcus	↑↓	[85,145]
	Bacteroidaceae	↓	[82]
	Faecalibacterium	↓	[82,143]
	*Faecalibacterium prausnitzii*	↓	[3]
	Ruminococcus	↑	[82]
	*Ruminococcus torques*	↑	[66]
	*Ruminococcus gnavus*	↑	[66]
	Methanobrevibacter	↑	[79,146]
	Akkermansia	↑	[79,143]
	*Akkermansia muciniphila*	↑	[3,141]
	Bifidobacterium	↓	[141,143]
	Pseudomonas	↑↓	[60]
	Mycoplana	↑↓	[60]
	Haemophilus	↑↓	[60]
	Blautia	↑↓	[3,60,143]
	*Blautia wexlerae*	↑	[66]
	*Blautia massiliensis*	↑	[66]
	Dorea	↑↓	[60]
	*Dysosmobacter welbionis*	↑	[66]
	*Flavonifractor plautii*	↑	[66]
	*Lawsonibacter phoceensis*	↑	[66]
	*Gordonibacter urolithinfaciens*	↑	[66]
	*Anaerobutyricum hallii*	↑	[66]
	*Pseudoflavonifractor capillosus*	↑	[66]
	*Anaerotruncus colihominis*	↑	[66]
	*Erysipelatoclostridium ramosum*	↑	[66]
	*Sellimonas intestinalis*	↑	[66]
	*Coprobacillus cateniformis*	↑	[66]
	*Ruthenibacterium lactatiformans*	↑	[3]
	*Eisenbergiella tayi*	↑	[3]
	*Bacteroides vulgatus*	↑	[141]
RR, Pediatric MS	Clostridium	↑↓	[84,140]
	Bilophila	↑↓	[84,140]
	Bacteroides 2 enterotype	↑	[139]
Pediatric MS	Escherichia	↑	[84]
	Shigella	↑	[84]
	*Eubacterium rectale*	↓	[84]
	Corynebacterium	↓	[84]
MS	*Akkermansia genus*	↑	[146]
	Firmicutes	↓	[143]
	Roseburia	↓	[143]
	Coprococcus	↓	[143]
	Butyricicoccus	↓	[143]
	Lachnospira	↓	[143]
	Dorea	↓	[143]
	Bacteroidetes	↑	[143]
	Ruminocococcus	↑	[143]
	Acinetobacter	↑	[142]
	*Parabacteroides distasonis*	↓	[142]

## Data Availability

No new data were created or analyzed in this study.

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
