# Peer review of "Microbiota-Driven Mechanisms in Multiple Sclerosis: Pathogenesis, Therapeutic Strategies, and Biomarker Potential"

_biology, 2025, doi:10.3390/biology14040435_

Round 1

Reviewer 1 Report

Comments and Suggestions for Authors

This submission discusses the valuable connection of the GBA with the risk and pathogenesis of MS. It covers several important points and provides substantial insights into the field involving the role of gut microbiota. However, this submission could greatly benefit from more revisions of the following issues:

1. The abstract is very dense and could be segmented into smaller sections so that the flow of the discussion can be understood better.
2. The section on the immunomodulatory effects of microbiota is very lengthy, and the main points of this section can be lost as the reader goes along. I suggest also breaking this down into smaller sections. One suggestion is to create another paragraph break for line 154, "Studies have demonstrated..." since B cells are now being discussed after T cells.
3. Consistency of terms: "gut microbiota" and "intestinal microbiota" are used interchangeably. I suggest sticking to one term throughout the review.
4. The section in lines 76-80 was italicized. Is there a reason why? If none, please remove the italicization. The same applies to "subsequent studies" and "status" in lines 227 and 231. Also "human and murine models" in line 212. 
5. Table 1 is informative but could be more visually appealing. For reference, please check this publication (Table 1): https://doi.org/10.3390/biology14030303. I also suggest using up and down arrows to indicate increased or decreased patterns and both up and down arrows for alternating patterns. However, this needs to be clearly defined in a table legend.

6. Acronyms and abbreviations should be clearly defined when first mentioned in the manuscript. Examples are EAE, Treg, and Th17. Please define all acronyms when they first appear in the manuscript: "experimental autoimmune encephalomyelitis (EAE)"
7. Several studies on EAE animal models were mentioned to correlate the findings in these models with MS. However, EAE models have several limitations which were not discussed. EAE is induced in these models through immunization with myelin proteins and does not accurately replicate the spontaneous occurrence of MS in human patients. MS is influenced by several genetic and environmental factors, which are not fully captured in EAE models. The manuscript can benefit from discussing the limitations of these models to guide future research and answer the remaining gaps.
8. Another limitation is that most of these studies rely on mice models, which have several biological differences in humans and may not reflect human neurological responses. 

9. Manuscript readability could also be improved through paragraph indentations.

10. Are there any specific mechanisms in which SCFAs help mitigate the symptoms of MS? Are these similar mechanisms involved in other neurodegenerative diseases?

11. Some species names are not italicized (i.e. "Akkermansia muciniphila" line 219, and all species mentioned in Table 1). Ranks above the genus level do not have to be italicized.

Author Response

This submission discusses the valuable connection of the GBA with the risk and pathogenesis of MS. It covers several important points and provides substantial insights into the field involving the role of gut microbiota. However, this submission could greatly benefit from more revisions of the following issues:

1. The abstract is very dense and could be segmented into smaller sections so that the flow of the discussion can be understood better.

Thank you for your valuable feedback. We have revised the abstract by segmenting it into smaller, more structured sections to improve readability and enhance the flow of discussion.

  1. The section on the immunomodulatory effects of microbiota is very lengthy, and the main points of this section can be lost as the reader goes along. I suggest also breaking this down into smaller sections. One suggestion is to create another paragraph break for line 154, "Studies have demonstrated..." since B cells are now being discussed after T cells.

We breaking this section into two parts as you suggested.

  1. Consistency of terms: "gut microbiota" and "intestinal microbiota" are used interchangeably. I suggest sticking to one term throughout the review.

In response to your suggestion, we have standardized the use of "gut microbiota" throughout the entire review and removed instances of "intestinal microbiota.

  1. The section in lines 76-80 was italicized. Is there a reason why? If none, please remove the italicization. The same applies to "subsequent studies" and "status" in lines 227 and 231. Also "human and murine models" in line 212. 

Thank you for your careful observation. The italicization in these lines, was unintended and does not serve a specific purpose in these sections. We have removed the italicization from these terms to maintain consistency and avoid unnecessary formatting.

  1. Table 1 is informative but could be more visually appealing. For reference, please check this publication (Table 1): https://doi.org/10.3390/biology14030303. I also suggest using up and down arrows to indicate increased or decreased patterns and both up and down arrows for alternating patterns. However, this needs to be clearly defined in a table legend.

In response, we have redesigned Table 1 to improve its visual clarity and structure.

  1. Acronyms and abbreviations should be clearly defined when first mentioned in the manuscript. Examples are EAE, Treg, and Th17. Please define all acronyms when they first appear in the manuscript: "experimental autoimmune encephalomyelitis (EAE)"

We have ensured that all abbreviations are fully spelled out the first time they appear in the text

  1. Several studies on EAE animal models were mentioned to correlate the findings in these models with MS. However, EAE models have several limitations which were not discussed. EAE is induced in these models through immunization with myelin proteins and does not accurately replicate the spontaneous occurrence of MS in human patients. MS is influenced by several genetic and environmental factors, which are not fully captured in EAE models. The manuscript can benefit from discussing the limitations of these models to guide future research and answer the remaining gaps.

We acknowledge that while EAE models are widely used to study MS, they have several limitations that should be discussed. In response, we have added a section addressing the limitations of EAE models. Specifically, we discuss that EAE is induced through immunization with myelin proteins, which does not fully replicate the spontaneous nature of MS in human patients.

  1. Another limitation is that most of these studies rely on mice models, which have several biological differences in humans and may not reflect human neurological responses.

We highlight that MS is influenced by complex genetic and environmental factors that are not entirely captured in EAE models.

  1. Manuscript readability could also be improved through paragraph indentations.

We have formatted the manuscript with consistent paragraph indentations to enhance clarity and ensure a more structured flow of information.

  1. Are there any specific mechanisms in which SCFAs help mitigate the symptoms of MS? Are these similar mechanisms involved in other neurodegenerative diseases?

SCFAs, such as butyrate and propionate, help regulate T-cell differentiation by promoting Treg cell expansion while suppressing pro-inflammatory Th1 and Th17 responses. They also contribute to maintaining blood-brain barrier integrity, reducing neuroinflammation, and modulating microglial activation, all of which are relevant to MS pathogenesis.

Furthermore, SCFAs play a role in other neurodegenerative diseases, such as Parkinson’s and Alzheimer’s disease. Similar to MS, SCFAs influence neuroinflammation and microglial function, regulate gut-brain axis interactions, and promote the production of anti-inflammatory cytokines.

  1. Some species names are not italicized (i.e. "Akkermansia muciniphila" line 219, and all species mentioned in Table 1). Ranks above the genus level do not have to be italicized.

We acknowledge the inconsistency in italicizing species names and have carefully revised the manuscript to ensure that all genus and species names, including Akkermansia muciniphila (line 235) and all species listed in Table 1, are correctly italicized. Additionally, we have verified that ranks above the genus level remain in standard formatting, as per scientific conventions.

Reviewer 2 Report

Comments and Suggestions for Authors

Dear Authors,

I have reviewed the review manuscript “Microbiota-Driven Mechanisms in Multiple Sclerosis: Pathogenesis, Therapeutic Strategies, and Biomarker Potential” by Nemati et al. In this review, the authors presented a comprehensive overview of interactions between the gut microbiota and multiple sclerosis. However, the manuscript needed major changes, and detailed comments are below.

Major comments

1) There are several related articles recently published that are missing from this manuscript (a few are added below). Authors need to cite those articles in the introduction at appropriate places to make the article stronger.

Microbial Metabolites in Multiple Sclerosis: Implications for Pathogenesis and Treatment https://www.frontiersin.org/journals/neuroscience/articles/10.3389/fnins.2022.885031/full

Microbial signatures and therapeutic strategies in neurodegenerative diseases DOI: 10.1016/j.biopha.2025.117905

Role of Gut Microbiota in Multiple Sclerosis and Potential Therapeutic Implications DOI:10.2174/1570159X19666210629145351

Targeting the gut to treat multiple sclerosis https://doi.org/10.1172/JCI143774.

The Role of the Gut Microbiome in Multiple Sclerosis Risk and Progression: Towards Characterization of the “MS Microbiome” https://doi.org/10.1007/s13311-017-0587-y

2) The authors needed to highlight the nobility of this article in the introduction to convey how this review article is different from other published articles.

3) It is important to add a methodology section that includes the sources and methods used to gather the information for the review, such as databases, search terms, or inclusion and exclusion criteria for the selected studies.

4) The English of the manuscript needs to be improved.

5) Formatting for references is not uniform.

Minor comments

Line 16 – line 17: - “have shown” can be written as “have been shown”

line 22: - “muciniphila to be associated”, can be written as “muciniphila was to be associated”.

line 27: -  remove “to be used”

line 30: - “microbiota composition in order to get at the precise mechanisms”. can be written as “microbiota composition to get the precise mechanisms”.

Line 149 :- “signals and shape” can be written as “signals and shapes”

Line 161 :- “In accordance with this fact,” can be written as “Following this fact”

Line 193:- transplantation appear to“Others can promote”

Line 244 :- First letter should be capitalized.

Line 266 :- “T regulatory cells and, as”. Remove and for clarification

Line 268:- “dysbiosis which leading to an” can be written as “dysbiosis leads to an”

Line 268:- “and presence ” can be written as “and the presence”

Line 366:- “ltimately,” should be written as “Ultimately”

Line 384:- “diets prior to immunization” can be written as “diets before immunization”

Line 417:- “increasing” can be written as “increases”

Line 469:- “downregulating” can be written as “downregulates”

Line 562:- “patient” can be written as “patients”

Line 691:- “research” can be written as “researches”

Line 696:- “epigenetic modifications such increased” can be written as “epigenetic modifications, such as increased”

Line 701:- “Investigating” can be written as “investigation”

Line 702:- “progress” can be written as “progress has been made”

Line 702:- “many parts still” can be written as “ many parts are still”

Line 703:- “explore” can be written as “explored”

Line 719:- “validate utility of FMT” can be written as “validate the utility of FMT”

Line 727:- “For” can be written as “for”

Line 727:- “therapeurtic” can be written as “therapeutic”

Line 727:- “MS and feasibility of ” can be written as “MS and the feasibility of ”

Line 755:- “transplantation appear to” can be written as “transplantation appears to”

  • At many places, “comma” is not used after the word “and”.
Comments on the Quality of English Language

The English of the manuscript needs to be improved.

Author Response

Dear Authors,

I have reviewed the review manuscript “Microbiota-Driven Mechanisms in Multiple Sclerosis: Pathogenesis, Therapeutic Strategies, and Biomarker Potential” by Nemati et al. In this review, the authors presented a comprehensive overview of interactions between the gut microbiota and multiple sclerosis. However, the manuscript needed major changes, and detailed comments are below.

Major comments

1) There are several related articles recently published that are missing from this manuscript (a few are added below). Authors need to cite those articles in the introduction at appropriate places to make the article stronger.

Microbial Metabolites in Multiple Sclerosis: Implications for Pathogenesis and Treatment https://www.frontiersin.org/journals/neuroscience/articles/10.3389/fnins.2022.885031/full

Microbial signatures and therapeutic strategies in neurodegenerative diseases DOI: 10.1016/j.biopha.2025.117905

Role of Gut Microbiota in Multiple Sclerosis and Potential Therapeutic Implications DOI:10.2174/1570159X19666210629145351

Targeting the gut to treat multiple sclerosis https://doi.org/10.1172/JCI143774.

The Role of the Gut Microbiome in Multiple Sclerosis Risk and Progression: Towards Characterization of the “MS Microbiome” https://doi.org/10.1007/s13311-017-0587-y

Thank you for your valuable suggestion. We have reviewed the suggested articles and integrated relevant citations into the article.

2) The authors needed to highlight the nobility of this article in the introduction to convey how this review article is different from other published articles.

We have revised the Introduction section to explicitly outline how our review provides a unique perspective.

3) It is important to add a methodology section that includes the sources and methods used to gather the information for the review, such as databases, search terms, or inclusion and exclusion criteria for the selected studies.

We added a paraghraph to clarify the sources and criteria used for selecting studies in this review. This section outlines the databases searched (e.g., PubMed, Scopus, Web of Science), the search terms employed (e.g., "gut microbiota AND multiple sclerosis," "microbiota-targeted therapies AND MS"), and the inclusion/exclusion criteria for study selection.

4) The English of the manuscript needs to be improved.

We have carefully revised the text to improve sentence structure, grammar, and flow.

5) Formatting for references is not uniform.

All references have been thoroughly reviewed and revised where necessary to ensure uniformity and adherence to the required citation style. The formatting is now consistent throughout the manuscript. 

Minor comments

We sincerely appreciate the reviewer’s detailed comments. All minor revisions have been carefully addressed.

Line 16 – line 17: - “have shown” can be written as “have been shown”

line 22: - “muciniphila to be associated”, can be written as “muciniphila was to be associated”.

line 27: -  remove “to be used”

line 30: - “microbiota composition in order to get at the precise mechanisms”. can be written as “microbiota composition to get the precise mechanisms”.

Line 149 :- “signals and shape” can be written as “signals and shapes”

Line 161 :- “In accordance with this fact,” can be written as “Following this fact”

Line 193:- transplantation appear to“Others can promote”

Line 244 :- First letter should be capitalized.

Line 266 :- “T regulatory cells and, as”. Remove and for clarification

Line 268:- “dysbiosis which leading to an” can be written as “dysbiosis leads to an”

Line 268:- “and presence ” can be written as “and the presence”

Line 366:- “ltimately,” should be written as “Ultimately”

Line 384:- “diets prior to immunization” can be written as “diets before immunization”

Line 417:- “increasing” can be written as “increases”

Line 469:- “downregulating” can be written as “downregulates”

Line 562:- “patient” can be written as “patients”

Line 691:- “research” can be written as “researches”

Line 696:- “epigenetic modifications such increased” can be written as “epigenetic modifications, such as increased”

Line 701:- “Investigating” can be written as “investigation”

Line 702:- “progress” can be written as “progress has been made”

Line 702:- “many parts still” can be written as “ many parts are still”

Line 703:- “explore” can be written as “explored”

Line 719:- “validate utility of FMT” can be written as “validate the utility of FMT”

Line 727:- “For” can be written as “for”

Line 727:- “therapeurtic” can be written as “therapeutic”

Line 727:- “MS and feasibility of ” can be written as “MS and the feasibility of ”

Line 755:- “transplantation appear to” can be written as “transplantation appears to”

Round 2

Reviewer 1 Report

Comments and Suggestions for Authors

The manuscript has been improved, however, there are still some issues that need to be addressed before this paper can be published. For spelling and formatting errors, please refer to my comments on English language:

  1. The table can still be improved. There are multiple instances of the duplicate entry "RRMS" in one column. I suggest grouping all taxa that correspond to RRMS together for easier visualization, and one column reads RRMS alone. You can refer to the attached PDF as an example.
  2. You included a section discussing the limitations of EAE animal models. However, many of the cited studies in your review used EAE models to recapitulate MS. Are EAE models currently the only model usable in MS research? Are there other animal models? If there are, please include them in the review. If not, please discuss this further in the limitations section.

Comments on the Quality of English Language

There are still several spelling and formatting errors that were not addressed.

  1. Line 43: mechanismsof
  2. 44: Conclusion > conclusion
  3. 149: regulatory T cells is more appropriate
  4. 217: Gut > gut
  5. 238: haveshown
  6. 258: enormous -> "numerous" may be a better term
  7. 265: F. prausnitzii is not italicized
  8. 275: strain > strains
  9. 284: monocolonization (remove dash)
  10. 291: remove "stimulating of the"
  11. 298: Treg was already defined. No need to define again.
  12. 300: Prevotella histocola is not italicized.
  13. 301: regulatory T-cells can be written as Tregs
  14. 657: All species mentioned were not italicized.
  15. 658: E. tayi was not italicized.
  16. 661: Bacteroides finegoldii (italicize please)
  17. 662: Roseburia faecis (italicize please)

    In the interest of time, I was not able to check all pages of the document, so I highly suggest that the authors carefully check all formatting and spelling again.

Author Response

The manuscript has been improved, however, there are still some issues that need to be addressed before this paper can be published. For spelling and formatting errors, please refer to my comments on English language:

Thank you for your valuable feedback. We appreciate your careful review and have thoroughly addressed all the remaining spelling and formatting issues. The manuscript has been carefully revised for spelling and formatting errors.

  1. The table can still be improved. There are multiple instances of the duplicate entry "RRMS" in one column. I suggest grouping all taxa that correspond to RRMS together for easier visualization, and one column reads RRMS alone. You can refer to the attached PDF as an example.

We have restructured the table to improve clarity and readability by grouping all taxa associated with RRMS together in a single section.

  1. You included a section discussing the limitations of EAE animal models. However, many of the cited studies in your review used EAE models to recapitulate MS. Are EAE models currently the only model usable in MS research? Are there other animal models? If there are, please include them in the review. If not, please discuss this further in the limitations section.

Thank you for your insightful comment. While Experimental Autoimmune Encephalomyelitis (EAE) is the most widely used model for studying MS due to its ability to mimic immune-mediated demyelination and neuroinflammation, it is not the only available model. In response to your suggestion, we have expanded the discussion in the limitations section to include other animal models of MS, such as:

The cuprizone model, which induces demyelination through toxic exposure, allowing the study of remyelination and neurodegeneration without direct immune system involvement.

The lysolecithin-induced demyelination model, which is useful for investigating myelin repair mechanisms and remyelination therapies.

Viral infection models, such as Theiler’s Murine Encephalomyelitis Virus (TMEV) model, which better reflects the progressive aspect of MS, including axonal damage and neurodegeneration.

These models provide complementary insights into different aspects of MS pathogenesis, including inflammation, neurodegeneration, and remyelination, addressing some limitations of the EAE model.

Comments on the Quality of English Language

There are still several spelling and formatting errors that were not addressed.

We sincerely appreciate the reviewer’s detailed comments. All spelling and formatting errors have been carefully addressed

Reviewer 2 Report

Comments and Suggestions for Authors

Authors made changes according to suggestions; now the manuscript can be accepted. 

Author Response

We sincerely appreciate the reviewer’s thorough evaluation and constructive feedback throughout the revision process. We are grateful for the positive assessment and the acceptance of our manuscript. Thank you for your valuable time and insightful suggestions, which have significantly improved the quality and clarity of our work.

Round 3

Reviewer 1 Report

Comments and Suggestions for Authors

All concerns were adequately addressed.